# Towards Trustworthy Amortized Bayesian Model Comparison

**Šimon Kucharský**
Department of Computational Statistics
Technical University Dortmund
Dortmund, Germany
simon.kucharsky@tu-dortmund.de

**Aayush Mishra**
Department of Computational Statistics
Technical University Dortmund
Dortmund, Germany
aayush.mishra@tu-dortmund.de

**Daniel Habermann**
Department of Computational Statistics
Technical University Dortmund
Dortmund, Germany
daniel.habermann@tu-dortmund.de

**Stefan T. Radev**
Department of Cognitive Science
Rensselaer Polytechnic Institute
Troy, NY, USA
radevs@rpi.edu

**Paul-Christian Bürkner**
Department of Computational Statistics
Technical University Dortmund
Dortmund, Germany
paul.buerkner@tu-dortmund.de

## Abstract

Amortized Bayesian model comparison (BMC) enables fast probabilistic ranking of models via simulation-based training of neural surrogates. However, the reliability of neural surrogates deteriorates when simulation models are misspecified—the very case where model comparison is most needed. Thus, we supplement simulation-based training with a *self-consistency* (SC) loss on unlabeled real data to improve BMC estimates under empirical distribution shifts. Using a numerical experiment and two case studies with real data, we compare amortized evidence estimates with and without SC against analytic or bridge sampling benchmarks. SC improves calibration under model misspecification when having access to analytic likelihoods. However, it offers limited gains with neural surrogate likelihoods, making it most practical for trustworthy BMC when likelihoods are exact.

## 1   Introduction

Bayesian model comparison (BMC) provides a formal method for ranking probabilistic models according to their compatibility with observed data [1]. However, BMC is often analytically or numerically intractable [2]. Amortized BMC circumvents this integration by training neural surrogates to approximate model-implied quantities, such as posterior model probabilities [3] or marginal likelihoods [4]. Despite their efficiency gains, neural networks are notoriously susceptible to out-of-distribution [OOD; 5] data. In amortized BMC, OOD data can arise from model misspecification [6, 7]: networks are trained on model simulations, but the real data can lie outside the generative scope of at least some candidate models. This creates a *conundrum*: neural estimates cannot be trusted for models that fit the data poorly, yet proper model comparison requires faithful estimates even for poorly fitting models. In this work, we study the reliability of amortized BMC under OOD

39th Conference on Neural Information Processing Systems (NeurIPS 2025) Workshop: Reliable ML from Unreliable Data.

conditions and ask: can we retain the efficiency of ABI while improving neural estimates when models are misspecified? To this end, we leverage the recently proposed self-consistency loss family [8–10] as a means to improve the calibration of Bayesian model comparison results.

## 2 Background

### 2.1 Bayesian model comparison (BMC)

Given data $y \in \mathcal{Y}$ and a set of candidate models $\mathcal{M} = \{M_1, \ldots, M_K\}$, posterior model probabilities (PMPs) follow Bayes' rule

$$p(M_k \mid y) = \frac{p(M_k) \, p(y \mid M_k)}{p(y)}, \qquad p(y) = \sum_{j=1}^{K} p(M_j) \, p(y \mid M_j). \tag{1}$$

Up to the prior model probabilities, models are ranked by their *marginal likelihoods* (model evidences) $p(y \mid M_k)$ [11]. Each marginal likelihood is an integral over the model parameters,

$$p(y \mid M_k) = \int p(\theta_k \mid M_k) \, p(y \mid \theta_k, M_k) \, d\theta_k, \tag{2}$$

which is intractable for most non-trivial models. Accurate approximations (e.g., bridge sampling, [12]) typically require: (i) the ability to evaluate the likelihood $p(y \mid \theta_k, M_k)$ and prior $p(\theta_k \mid M_k)$; (ii) many approximately independent posterior draws $\theta_k^{(s)} \sim p(\theta_k \mid y, M_k)$ [13, 14], motivating simulation-based inference (SBI) for estimating $p(y \mid M_k)$ or $p(M_k \mid y)$.

### 2.2 Amortized Bayesian inference (ABI)

ABI can be viewed as learning a conditional distribution $q_\phi(z \mid y)$ from simulations $(z^{(n)}, y^{(n)}) \sim p(z, y)$. We train $q_\phi$ by minimizing a scoring rule,

$$\hat{\phi} = \arg\min_{\phi} \ \frac{1}{N} \sum_{n=1}^{N} S\big(q_\phi(\cdot \mid y^{(n)}), \, z^{(n)}\big), \tag{3}$$

which ensures $q_\phi(z \mid y) \to p(z \mid y)$ as $N \to \infty$ if $q_\phi$ is sufficiently expressive and $S$ is strictly proper (e.g., the log score $-\log q_\phi(z^{(n)} \mid y^{(n)})$, [15]).

**Instantiations** *(i) Parameter posterior (fixed model $M_k$).* Let $Z = \Theta_k$, draw $(\theta_k^{(n)}, y^{(n)}) \sim p(\theta_k, y \mid M_k)$, and train a generative neural network (e.g., a coupling flow) $q_\phi(\theta_k \mid y) \approx p(\theta_k \mid y, M_k)$. *(ii) Model posterior (model comparison).* Let $Z = M$ with prior model probabilities $p(M)$, draw $M^{(n)} \sim p(M)$ and $y^{(n)} \sim p(y \mid M^{(n)})$, and train a classifier $q_\phi(M \mid y) \approx p(M \mid y)$.

**Self-consistency (SC)** Trained on finite synthetic data, the networks can be arbitrarily wrong if applied to OOD data [6, 7]. One option to make inferences more trustworthy for empirical data is adding a *self-consistency* loss to the optimization objective,

$$\hat{\phi} = \arg\min_{\phi} \ \frac{1}{N} \sum_{n=1}^{N} S\big(q_\phi(\cdot \mid y^{(n)}), \, z^{(n)}\big) + \lambda_{\text{SC}} \frac{1}{M} \sum_{m=1}^{M} C\left[\frac{p(z)p(y^{(m)} \mid z)}{q_\phi(z \mid y^{(m)})}\right], \tag{4}$$

leveraging unlabeled datasets $y^{(m)}$. A suitable choice of $C$ is

$$C\left[\frac{p(z)p(y^{(m)} \mid z)}{q_\phi(z \mid y^{(m)})}\right] = \text{Var}_{z^* \sim p_c(z)}\Big[\log p(z^*) + \log p(y^{(m)} \mid z^*) - \log q_\phi(z^* \mid y^{(m)})\Big], \tag{5}$$

with a proposal distribution $p_c$ typically set to the current approximate posterior $q_\phi$ [10].

## 3 Method

**Estimating the marginal likelihood**    We estimate the log marginal likelihood $\log p(y \mid M_k)$ by rearranging the Bayes' theorem and replacing posterior density with a neural surrogate $q_\phi$,

$$\log p(y \mid M_k) \approx \log p(\theta_k^* \mid M_k) + \log p(y \mid \theta_k^*, M_k) - \log q_\phi(\theta_k^* \mid y, M_k). \tag{6}$$

If $q_\phi$ is converged, the estimate is correct for any $\theta_k^* \in \Theta_k$. In practice, we average the estimate over multiple draws $\theta_k^* \sim q_\phi(\theta_k \mid y)$. For models with intractable likelihoods, we may train a surrogate likelihood $q_\psi(y \mid \theta_k, M_k)$ in tandem with the posterior [4]; however, this lifts up the strict properness of SC losses [10]. Here, we investigate whether SC improves the estimation of marginal likelihoods (i.e., evidences) for OOD data, and whether it helps estimation even with an approximate likelihood. See Appendix A for more details about estimating the marginal likelihood and the SC loss.

**Estimating posterior model probabilities**    Posterior model probabilities can be estimated via a classifier trained with a cross-entropy loss [3]. Adding the SC loss, Eq. 3 becomes:

$$\hat{\phi} = \arg\min_\phi \frac{1}{N} \sum_{n=1}^{N} \log q_\phi\big(M_k^{(n)} \mid y^{(n)}\big) +$$

$$\lambda_{\text{SC}} \frac{1}{M} \sum_{m=1}^{M} \text{Var}_{M_k \sim p_c(M_k)} \Big[ \log p\big(M_k\big) + \log p\big(y^{(m)} \mid M_k\big) - q_\phi\big(M_k \mid y^{(m)}\big) \Big]. \tag{7}$$

The SC loss requires access to the marginals $p(y \mid M_k)$. Instead, one might approximate these with a surrogate evidence network $q_\gamma(y \mid M_k)$; in that case, the SC loss is not strictly proper either [10]. For the sake of brevity, we focus this report on the estimating marginal likelihood approach.

## 4 Related Work

The following brief survey of related work focuses on neural methods targeting BMC in SBI. Amortized BMC via classifiers was systematically studied by Radev et al. [3] and has been applied across various domains [e.g., 16–18]. Elsemüller et al. [19] extended the approach to hierarchical models; Schröder and Macke [20] enabled joint parameter estimation and model comparison. Jeffrey and Wandelt [21] surveyed loss functions for estimating PMPs and Bayes factors (BFs). Elsemüller et al. [22] proposed deep ensembles for detecting OOD data. The JANA method allowed estimating the evidence from a joint approximation of the posterior and likelihood [4]. Spurio Mancini et al. [23] proposed to "post-process" posterior samples using a learned harmonic mean estimator. Srinivasan et al. [24] used normalizing flows to estimate the evidence and its uncertainty from posterior samples. Schmitt et al. [8] showed that the SC loss improves evidence estimation on well-specified models. None of these works considered *repairing* the "extrapolation bias" [7] of BMC quantities in the face of model misspecification. We adapt the SC loss of Schmitt et al. [8] and test its potential for reliable BMC under model misspecification.

## 5 Experiments

### 5.1 Experiment 1: Multivariate Gaussian

We start with a $D$-dimensional Gaussian location model with $N$ IID observations, with prior $\boldsymbol{\mu} \sim \mathcal{N}(\mathbf{0}, \sigma_\mu^2 \mathbf{I}_D)$ and likelihood $\mathbf{y}_i \sim \mathcal{N}(\boldsymbol{\mu}, \sigma_y^2 \mathbf{I}_D)$ for $i = 1, \ldots, N$. During training we use either a fixed prior scale ($\sigma_\mu^2 = 1$) or a randomized prior scale with $\log \sigma_\mu^2 \sim \text{Uniform}(-3, 3)$; the latter is passed as an input encoding different models with varying prior concentration. We use either neural posterior estimation (NPE) or neural likelihood and posterior estimation (NLPE). For the SC loss, we use $M = 32$ datasets drawn from $\mathcal{N}(\mathbf{5}, \sigma_y^2 \mathbf{I}_D)$, *i.e.*, a mean shift away from the prior location. See Appendix B for details about the network architectures.

**Results**    We compare log marginal likelihoods from NPE, NPE+SC, NLPE, and NLPE+SC to the analytic ground truth. Figure 1 (top row) reports the case $N = 100$, $D = 10$, with test means $\boldsymbol{\mu} \in \{\mathbf{0}, \mathbf{5}, \mathbf{8}\}$. (i) **NPE:** In-distribution ($\boldsymbol{\mu} = \mathbf{0}$) estimates are essentially exact. Under OOD shift

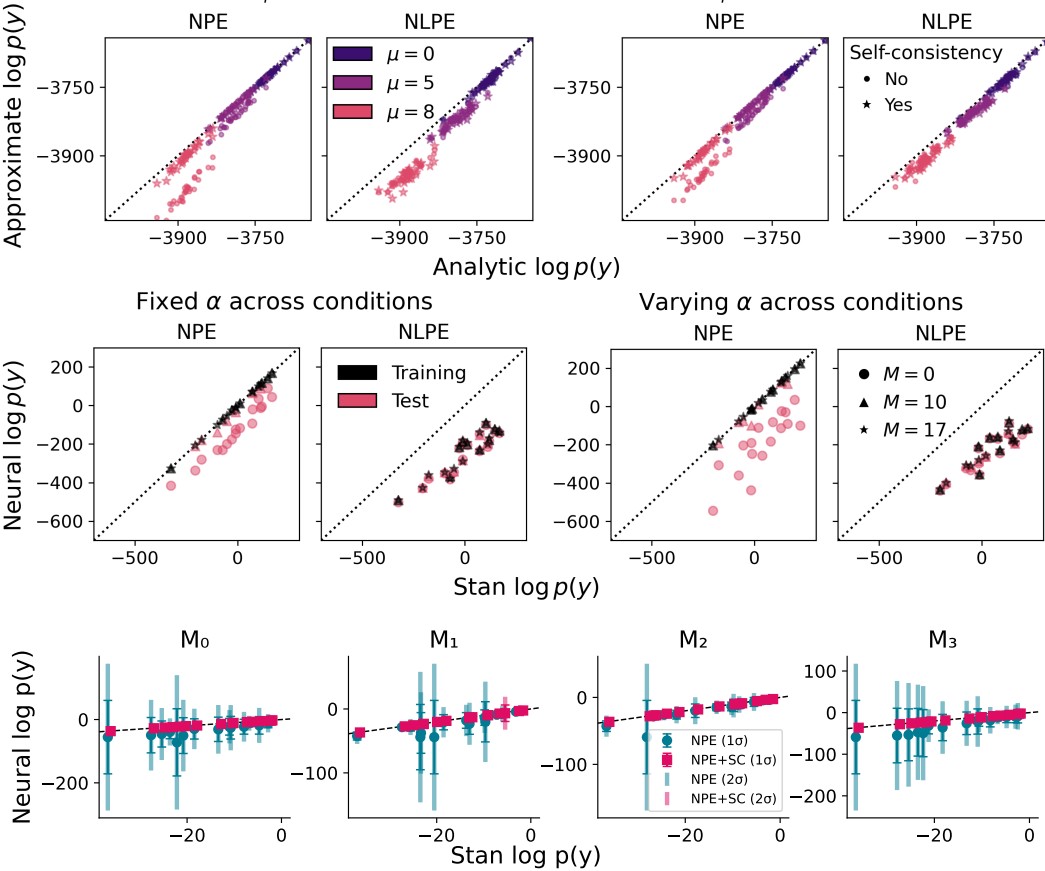

Figure 1: Log marginal likelihood estimates comparing neural estimators (*y*-axis) and the gold standard (*x*-axis) for the Gaussian (**Experiment 1**, *top row*), racing diffusion (**Experiment 2**, *middle row*), and air passenger traffic (**Experiment 3**, *bottom row*) experiments.

matching the SC data ($\mu = 5$), SC markedly improves calibration; at a larger shift ($\mu = 8$), SC still helps but does not fully close the gap. (ii) **NLPE:** SC loss yields little gain due to not being strictly proper when both posterior and likelihood are approximate [10]. Expanding the training context across prior scales ($\log \sigma_\mu^2 \sim \mathrm{Uniform}(-3, 3)$) improves OOD estimation, though less markedly.

### 5.2 Experiment 2: Racing diffusion models of decision making

Next, we analyze real data from the lexical decision task of Wagenmakers et al. [25] (Experiment 1), where 17 participants judged letter strings as words or non-words under speed or accuracy instructions. Decision and response times are modeled with the racing diffusion model [26], where two noisy accumulators compete against each other in a race towards a threshold $\alpha$ to trigger its response. Under the null model, $\alpha$ is shared across conditions; under the alternative, it varies by condition. We use NPE and NLPE. For the SC loss, we use the data from $M \in \{0, 10, 17\}$ participants ($M = 0$ corresponds to no SC training). See Appendix C for more details on the models and network architectures.

**Results** We compare log marginal likelihood from all 17 participants using NPE, NPE+SC, NLPE, NLPE+SC to bridge sampling [12] with NUTS, as implemented in Stan [27, 28] as the gold standard. Figure 1 (bottom row) shows the results. (i) **NPE:** Without SC, the estimates are relatively far away from the gold standard. With SC, estimates become very close to the gold standard, even for data sets that were not included in the SC loss. (ii) **NLPE:** SC does not improve the estimates.

### 5.3 Experiment 3: Air passenger traffic forecasting

Finally, we apply the SC loss to compare different models that predict trends in European air passenger traffic data [29–31]. This case study and has been previously used to demonstrate benefits of the SC loss for parameter estimation [10]. The annual time series data on air passenger departures from 15 European countries to the USA in 2004 – 2019 were fitted by four first-order autoregressive models. We train NPE with a small simulation budget and utilize the data from $M \in \{4, 8, 15\}$ countries for the SC loss. The test dataset comprises data from all 15 countries. See Appendix D for more details about the model, network architectures and training, along with additional results.

**Results**   We estimated log marginal likelihood using NPE, NPE+SC, and bridge sampling [12] with Stan [27, 28] as the gold standard. Figure 1 (bottom row) compares the neural estimates against bridge sampling. Without SC, the estimates deviate substantially from the gold standard and exhibit high variance. With SC, estimates exhibit small variance and high agreement with the gold standard.

## 6   Conclusion

Our initial results suggest that self-consistency can substantially improve amortized Bayesian model comparison when likelihoods are tractable, yielding better calibrated marginal likelihood estimates for OOD data resulting from model misspecification. However, the benefits of self-consistency diminish when using surrogate likelihoods, highlighting the need for more research on expanding its scope.

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

# Appendix

## A   The marginal likelihood and self-consistency

We start with the Bayes' theorem for parameter posterior given data,

$$p(\theta_k \mid y, M_k) = \frac{p(\theta_k \mid M_k) \times p(y \mid \theta_k, M_k)}{p(y \mid M_k)}, \tag{8}$$

where the marginal likelihood is defined as

$$p(y \mid M_k) = \int p(\theta_k \mid M_k)\, p(y \mid \theta_k, M_k)\, d\theta_k, \tag{9}$$

an integral over the model parameters that is typically intractable. However, by rearranging the Bayes' theorem, we obtain an identity that expresses the marginal likelihood in terms of prior, likelihood, and posterior,

$$p(y \mid M_k) = \frac{p(\theta_k \mid M_k) \times p(y \mid \theta_k, M_k)}{p(\theta_k \mid y, M_k)}. \tag{10}$$

If the true posterior $p(\theta_k \mid y, M_k)$ were available, this equality would hold exactly for any $\theta_k \in \Theta_k$. In practice, however, the posterior is intractable, which motivates replacing it with a neural surrogate.

**Marginal likelihood estimated with NPE**   We use a generative neural network $q_\phi(\theta_k \mid y, M_k)$ to *approximate* the posterior $p(\theta_k \mid y, M_k)$, an approach known as neural posterior estimation (NPE). Substituting $q_\phi$ into Equation 10 and taking the log yields

$$\log p(y \mid M_k) \approx \log p(\theta_k \mid M_k) + \log p(y \mid \theta_k, M_k) - \log q_\phi(\theta_k \mid y, M_k). \tag{11}$$

In principle, if $q_\phi$ were exact, the estimate would be correct for any $\theta_k \in \Theta_k$. In practice, deviations from the true posterior may make estimates based on a single $\theta_k$ unreliable. To mitigate this, we compute its expectation over the surrogate posterior with a Monte Carlo estimate,

$$\log p(y \mid M_k) \approx$$
$$\frac{1}{S} \sum_{s=1}^{S} \big[ \log p(\tilde{\theta}_k^{(s)} \mid M_k) + \log p(y \mid \tilde{\theta}_k^{(s)}, M_k) - \log q_\phi(\tilde{\theta}_k^{(s)} \mid y, M_k) \big], \tag{12}$$

with $S$ samples drawn from the posterior surrogate $\tilde{\theta}_k^{(s)} \sim q_\phi(\theta_k \mid y, M_k)$.

**Self-consistency**   The self-consistency (SC) objective builds on the same estimator of marginal likelihood but emphasizes the property of invariance with respect to parameter values: if $q_\phi$ is close to the true posterior, the marginal likelihood estimate should be nearly the same regardless of which parameter value is used. Here, we define the SC loss to minimize the variance of the marginal likelihood estimator with respect to the posterior,

$$\mathrm{Var}_{\tilde{\theta}_k \sim q_\phi(\theta_k \mid y, M_k)} \big[ \log p(\tilde{\theta}_k \mid M_k) + \log p(y \mid \tilde{\theta}_k, M_k) - \log q_\phi(\tilde{\theta}_k \mid y, M_k) \big]. \tag{13}$$

During training, this variance is approximated with a finite number of posterior samples.

**Marginal likelihood estimated with NLPE**   In cases where the likelihood $p(y \mid \theta_k, M_k)$ is unavailable (e.g, implicit simulators), we may jointly approximate both likelihood and posterior with neural surrogates (NLPE), $q_\psi(y \mid \theta_k, M_k)$ and $q_\phi(\theta_k \mid y, M_k)$, respectively. The marginal likelihood estimate becomes

$$\log p(y \mid M_k) \approx \log p(\theta_k \mid M_k) + \log q_\psi(y \mid \theta_k, M_k) - \log q_\phi(\theta_k \mid y, M_k). \tag{14}$$

# B  Details of the multivariate Gaussian experiment

The generative model with $D$ dimensions and $N$ IID observations is as follows,

$$\begin{aligned}
\boldsymbol{\mu} &\sim \mathcal{N}(\mathbf{0},\, \sigma_\mu^2 \mathbf{I}_D) \\
\mathbf{y}_i &\sim \mathcal{N}(\boldsymbol{\mu},\, \sigma_y^2 \mathbf{I}_D) \quad \text{for } i = 1, \ldots, N.
\end{aligned} \tag{15}$$

Throughout all experiments, we hold $\sigma_\mu^2 = N$ so that the total amount of information about $\boldsymbol{\mu}$ remains approximately constant regardless of $N$. During training we use either a fixed prior scale ($\sigma_\mu^2 = 1$) or a randomized prior scale with $\log \sigma_\mu^2 \sim \mathrm{Uniform}(-3, 3)$; the latter is passed as an input encoding different models with varying prior concentration.

The neural posterior estimators and neural likelihood estimators were both implemented as normalizing flows [32] with six affine coupling layers, each consisting of two dense layers containing 256 hidden units and mish activations [33]. A DeepSet [34] architecture was used as a summary network, consisting of two equivariant modules each with 2 dense layers with 64 hidden units and an invariant module with 2 dense layers with 64 hidden units both in the inner and outer layer, followed by additional invariant module with two dense layers with 64 hidden units each, and an output layer of 30 units. All layers except for the output layer used SILU activations. We used the MMD loss on the output of the summary network [6].

We use either neural posterior estimation (NPE) or simultaneous neural likelihood and posterior estimation (NLPE) with a standard SBI loss trained online (50 epochs, 64 steps/epoch, batch size 64). Because the SC loss tends to be unstable for randomly initiated networks, the SC weight is warm-started: $\lambda_{\mathrm{SC}} = 0$ for epochs 1–20, increases linearly from 0 to 1 over epochs 21–30, and is held at 1 thereafter. We used $S = 16$ posterior samples to approximate the variance of log marginal likelihood for the SC loss (Equation 5). During testing, we used $S = 128$ posterior samples to approximate the log marginal likelihood.

**Additional results**  We trained NPE and NLPE with and without SC loss, based on different sample sizes $N \in \{10, 100\}$ and number of dimensions $D \in \{1, 10\}$. For each combination of $N$ and $D$, we generated $K = 512$ data sets with $\boldsymbol{\mu} \in \{\mathbf{0}, \mathbf{5}, \mathbf{8}\}$. To asses the closeness of the approximation to the analytic solution $\log p(y_k)$, we calculated the root mean squared error

$$\mathrm{RMSE} = \sqrt{\frac{1}{K} \sum_{k=1}^{K} \Big( \log q(y_k) - \log p(y_k) \Big)^2}. \tag{16}$$

To ascertain whether the SC loss improves marginal likelihood estimation, we computed $\Delta\mathrm{RMSE} = \mathrm{RMSE}_{\mathrm{SC}} - \mathrm{RMSE}_{\mathrm{no\ SC}}$; negative values indicate improvement when using SC loss. The results are summarised in Table 1. (i) **NPE**: SC improves estimation across the board with the exception of in-distribution ($\boldsymbol{\mu} = \mathbf{0}$) where the estimation is already precise. (ii) **NLPE**: Benefits of SC are inconsistent.

Table 1: Additional results for the multivariate Gaussian experiment. $\Delta$RMSE of log marginal likelihood estimates: negative values show improvement with SC.

| | $N$ | $D$ | $\sigma_\mu = 1$ | | | $\log \sigma_\mu^2 \sim \mathrm{Uniform}(-3,3)$ | | |
| --- | --- | --- | --- | --- | --- | --- | --- | --- |
| | | | $\mu = 0$ | $\mu = 5$ | $\mu = 8$ | $\mu = 0$ | $\mu = 5$ | $\mu = 8$ |
| NPE | 10 | 1 | 0.001 | **-0.025** | **-0.171** | 0.001 | **-0.041** | **-0.376** |
| | | 10 | **-0.693** | **-28.349** | **-81.807** | **-0.187** | **-38.079** | **-100.218** |
| | 100 | 1 | 0.002 | **-0.039** | **-0.238** | 0.003 | **-0.553** | **-2.190** |
| | | 10 | **-0.002** | **-24.100** | **-77.127** | **-0.995** | **-21.927** | **-50.971** |
| NLPE | 10 | 1 | **-0.012** | 0.017 | **-0.260** | 0.006 | **-0.067** | **-0.122** |
| | | 10 | 0.120 | **-11.193** | 43.329 | **-0.049** | **-10.121** | **-16.926** |
| | 100 | 1 | 0.049 | **-0.020** | **-0.193** | 0.046 | **-0.628** | **-1.681** |
| | | 10 | **-0.093** | 0.656 | 3.790 | 0.637 | 3.595 | 6.833 |

## C  Details of the racing diffusion experiment

The racing diffusion model [26] of decision making assumes that for each response (in this case, correct/incorrect) there is one noisy evidence accumulator. The accumulators start at zero and evolve as

$$X_t = \nu \, \mathrm{d}t + \sigma \, \mathrm{d}W_t, \quad X_0 = 0, \tag{17}$$

with $W_t$ a standard Wiener process. A response occurs when an accumulator reaches the decision boundary $\alpha$. The final response time is the time it took the first accumulator to reach the decision boundary, plus a constant non-decision time, $t_0$. The model generates response times ($rt$, in seconds) and accuracy ($d$, correct/incorrect) for each trial. For the purposes of training, we recoded the data such that incorrect responses were coded as negative value of the $rt$ for that trial. This allowed us to condition the networks on one variable, representing speed and accuracy of the response.

The experiment alternated between speed and accuracy condition blocks; during each block, participants were either instructed to be as fast, or instructed to respond accurately. Each block consisted of 96 trials. For the purposes of this experiment, we selected the first four blocks (two blocks per condition) for each participant, each dataset therefore contains 384 trials (192 per condition).

Under the null model, $\alpha$ is shared across conditions,

$$\begin{aligned} M_0 &: \text{Fixed } \alpha \text{ across conditions} \\ \log \alpha &\sim \mathcal{N}(0, 0.5), \end{aligned} \tag{18}$$

and under the alternative model $\alpha$ varies by condition (i.e., speed and accuracy blocks have different decision thresholds),

$$\begin{aligned} M_1 &: \text{Varying } \alpha \text{ across conditions} \\ \log \alpha_{\text{speed}} &\sim \mathcal{N}(0, 0.5) \\ \log \alpha_{\text{accuracy}} &\sim \mathcal{N}(0, 0.5). \end{aligned} \tag{19}$$

The alternative model corresponds to the hypothesis that participants respond to instructions by modifying their response caution (how much evidence they require before making a decision).

Under both models the following priors were used on the nuisance parameters,

$$\begin{aligned} \log \nu_{\text{correct}} &\sim \mathcal{N}(0, 0.5) \\ \log \nu_{\text{incorrect}} &\sim \mathcal{N}(0, 0.5) \\ \text{logit } \tau &\sim \mathcal{N}(0, 1), \end{aligned} \tag{20}$$

with $\sigma = 1$. The relationship between parameter $\tau$ and non-decision time $t_0$ is as follows:

$$t_0 = \min(rt) \frac{\tau}{1 - \tau}. \tag{21}$$

Using this re-parametrization allowed us to estimate the parameter $\text{logit } \tau$ on the unconstrained logit scale, while $t_0$ is bounded between 0 and $\min(rt)$.

The neural posterior estimators and neural likelihood estimators were both implemented as normalizing flows [32] with six affine coupling layers, each consisting of two dense layers containing 256 hidden units and mish activations [33]. A DeepSet [34] architecture was used as a summary network, consisting of two equivariant modules each with 2 dense layers with 64 hidden units and an invariant module with 2 dense layers with 32 hidden units both in the inner and outer layer, followed by additional invariant module with two dense layers with 32 hidden units each, and an output layer of 24 units. All layers except for the output layer used SILU activations.

We train the NPE/NLPE networks in an online training regime (300 epochs, 64 steps/epoch, batch size 64). Because the SC loss tends to be unstable for randomly initiated networks (especially when both posterior and likelihood are approximate), the SC weight is warm-started: $\lambda_{\mathrm{SC}} = 0$ for epochs 1–100, increases linearly from 0 to 1 over epochs 101–200, and is held at 1 thereafter. We used $S = 16$ posterior samples to approximate the variance of log marginal likelihood for the SC loss (Equation 5). During testing, we used $S = 128$ posterior samples to approximate the log marginal likelihood.

## D  Details of the air passenger traffic experiment

We fitted the following first order autoregressive models:

$$M_0: \quad y_{j,t+1} \sim \mathcal{N}(\alpha_j + y_{j,t}\beta_j + u_{j,t}\gamma_j + w_{j,t}\delta_j, \sigma_j), \tag{22a}$$

$$M_1: \quad y_{j,t+1} \sim \mathcal{N}(\alpha_j + y_{j,t}\beta_j + w_{j,t}\delta_j, \sigma_j), \tag{22b}$$

$$M_2: \quad y_{j,t+1} \sim \mathcal{N}(\alpha_j + y_{j,t}\beta_j + u_{j,t}\gamma_j, \sigma_j), \tag{22c}$$

$$M_3: \quad y_{j,t+1} \sim \mathcal{N}(\alpha_j + u_{j,t}\gamma_j + w_{j,t}\delta_j, \sigma_j), \tag{22d}$$

where the target quantity $y_{j,t+1}$ represents the change in air passenger traffic for country $j$ from time $t$ to $t+1$. We incorporate two additional predictors: $u_{j,t}$ denotes the annual household debt of country $j$ at time $t$, expressed as a percentage of gross domestic product (GDP), and $w_{j,t}$ denotes the real GDP per capita. We consider the full model (Eq. 22a) and three reduced models obtained by removing one parameter each (Eqs. 22b–22d).

For the air traffic model described above we assign independent prior distributions to the parameters similar to Mishra et al. [10]:

$$\alpha_j \sim \mathcal{N}(0, 0.5) \qquad\qquad \beta_j \sim \mathcal{N}(0, 0.2)$$
$$\gamma_j \sim \mathcal{N}(0, 0.5) \qquad\qquad \delta_j \sim \mathcal{N}(0, 0.5)$$
$$\log(\sigma_j) \sim \mathcal{N}(-1, 0.5)$$

The parameters $\alpha_j$ represent country-specific intercepts, $\beta_j$ are the autoregressive coefficients, $\gamma_j$ are the regression coefficients for household debt, $\delta_j$ are the regression coefficients for GDP per capita, and $\sigma_j$ is the standard deviation of the noise term.

The neural posterior estimators are implemented as neural spline flows [35, 36] with six coupling layers, consisting of two dense layers each containing 256 hidden units and exponential linear unit activations. We apply L2 weight regularization with coefficient $\gamma = 10^{-3}$, a dropout rate of 5%, and a multivariate standard Gaussian latent space. These architectural choices are kept identical for training both NPE and NPE+SC.

The summary network consists of a long short-term memory layer producing 64-dimensional outputs, followed by two fully connected layers with 256 and 64 units, respectively. Both the inference and summary networks are trained jointly for 40 epochs using the Adam optimizer, a batch size of 32, and a learning rate of $5 \times 10^{-4}$. These hyperparameters were kept the same across all the four models (Equations 22a, 22b, 22c, 22d) for both NPE and NPE+SC.

We train NPE with a small simulation budget of $N = 1024$ simulations and utilize the unlabeled data from $M \in \{4, 8, 15\}$ countries for the SC loss. The offline training was done for 40 epochs with a batch size of 32.

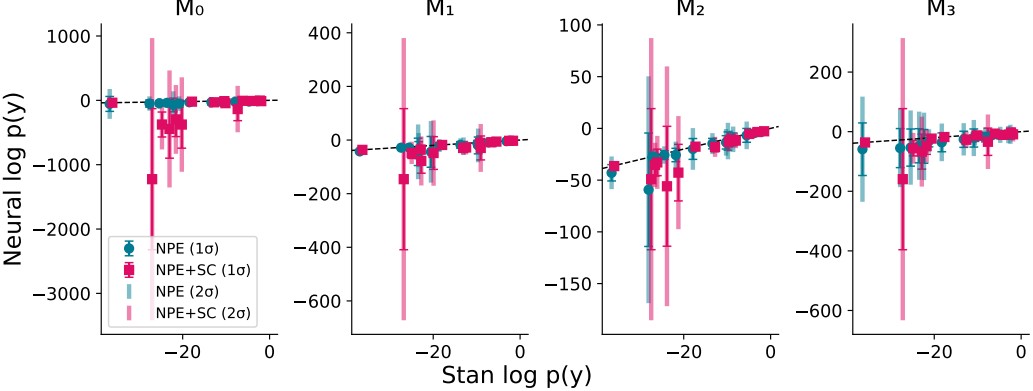

Figure 2: Comparison of the estimated $\log p(y)$ from NPE+SC and NPE against the gold-standard bridge-sampling results. The dataset from $M = 4$ countries was used to evaluate the SC loss and $\log p(y)$ was estimated using 256 Monte Carlo samples.

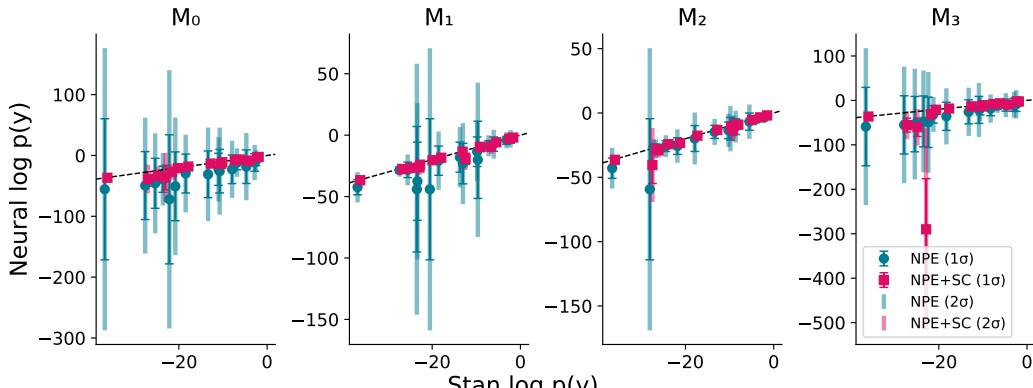

Figure 3: Comparison of the estimated $\log p(y)$ from NPE+SC and NPE against the gold-standard bridge-sampling results. The dataset from $M = 8$ countries was used to evaluate the SC loss and $\log p(y)$ was estimated using 256 Monte Carlo samples.

