# OpenReview forum: "Towards Trustworthy Amortized Bayesian Model Comparison"
_NeurIPS.cc/2025/Workshop/Reliable_ML — NeurIPS 2025 - Reliable ML Workshop_

### Official Review · Reviewer_YoGh · 2025-09-20
**Sound investigation on self-consistency for amortized Bayesian model comparison under model misspecification**

**Rating:** 7
**Confidence:** 2

**Review:**

**Summary**

This work investigates self-consistency for the reliability of amortized Bayesian model comparison (BMC) under model misspecification, with the finding being that self-consistency can improve amortized BMC when likelihoods are tractable, and the benefits of which diminish with surrogate likelihoods.

**Strengths**

Overall this work is a sound investigation and contribution to amortized BMC under model misspecification.

1. The problem to be addressed is very well-motivated and articulated.
2. The empirical evaluation is diverse and sound with 3 different and representative task datasets.
3. The positioning of this work is well done with a thorough related work analysis.

**Weaknesses**

1. clarity on result presentation could be improved. Though marked explicitly, the figures for each experiment result could be arranged in a clearer layout, e.g. present all results in a single figure with 3 panels, or present each result in a single figure.

**Suggestions**

1. Line 109 - *This case study and…* remove ‘and’.